# Kidney Transplant-Associated Viral Infection Rates and Outcomes in a Single-Centre Cohort

**DOI:** 10.3390/v14112406

**Published:** 2022-10-29

**Authors:** Kairi Pullerits, Shona Garland, Sharmilee Rengarajan, Malcolm Guiver, Rajkumar Chinnadurai, Rachel J. Middleton, Chukwuma A. Chukwu, Philip A. Kalra

**Affiliations:** 1Department of Undergraduate Medicine, University of Manchester, Oxford Road, Manchester M13 9PL, UK; 2Department of Nephrology, Salford Royal Hospital, Northern Care Alliance NHS Foundation Trust, Salford M6 8HD, UK; 3Department of Virology Manchester, University NHS Foundation Trust, Oxford Road, Manchester M13 9WL, UK; 4Faculty of Biology, Medicine and Health, Division of Cardiovascular Medicine, University of Manchester, Oxford Road, Manchester M13 9PL, UK

**Keywords:** kidney transplantation, CMV viremia, EBV viremia, BKV viremia, JCV viremia

## Abstract

Background: Opportunistic infections remain a significant cause of morbidity and mortality after kidney transplantation. This retrospective cohort study aimed to assess the incidence and predictors of post-transplant DNA virus infections (CMV, EBV, BKV and JCV infections) in kidney transplant recipients (KTR) at a single tertiary centre and evaluate their impact on graft outcomes. Methods: KTR transplanted between 2000 and 2021 were evaluated. Multivariate logistic regression analysis and Cox proportional hazard analyses were used to identify factors associated with DNA virus infections and their impact on allograft outcomes respectively. A sub-analysis of individual viral infections was also conducted to describe the pattern, timing, interventions, and outcomes of individual infections. Results: Data from 962 recipients were evaluated (Mean age 47.3 ± 15 years, 62% male, 81% white). 30% of recipients (288/962) had infection(s) by one or more of the DNA viruses. Individually, CMV, EBV, BKV and JCV viruses were diagnosed in 13.8%. 11.3%, 8.9% and 4.4% of recipients respectively. Factors associated with increased risk of post-transplant DNA virus infection included recipient female gender, higher number of HLA mismatch, lower baseline estimated glomerular filtration rate (eGFR), CMV seropositive donor, maintenance with cyclosporin (rather than tacrolimus) and higher number of maintenance immunosuppressive medications. The slope of eGFR decline was steeper in recipients with a history of DNA virus infection irrespective of the virus type. Further, GFR declined faster with an increasing number of different viral infections. Death-censored graft loss adjusted for age, gender, total HLA mismatch, baseline eGFR and acute rejection was significantly higher in recipients with a history of DNA virus infection than those without infection (adjusted hazard ratio (aHR, 1.74, 95% CI, 1.08–2.80)). In contrast, dialysis-free survival did not differ between the two groups of recipients (aHR, 1.13, 95% CI, 0.88–1.47). Conclusion: Post-transplant DNA viral infection is associated with a higher risk of allograft loss. Careful management of immunosuppression and close surveillance of at-risk recipients may improve graft outcomes.

## 1. Introduction

Patients who undergo solid organ transplantation have a significantly higher risk of developing infections and infection-related complications than the general population [1]. Infection is the leading non-cardiovascular cause of death accounting for 15–20% of all deaths in kidney transplant recipients (KTR) [2].

Pathogens that commonly infect transplant recipients include viruses, bacteria, and protozoa. Of these, viruses pose the biggest threat [3]. Viruses commonly associated with kidney transplantation include CMV, Epstein–Barr virus (EBV), BK virus (BKV), and John Cunningham virus (JCV). These viruses are highly prevalent in the healthy population, existing in latent asymptomatic states. The reported prevalence rates of CMV, EBV, BKV and JCV are 60%, 90%, 80%, and 80%, respectively, [3,4]. In immunosuppressed individuals, they can reactivate to cause a range of systemic diseases including allograft dysfunction. The first six months after transplantation are associated with the highest risk of community-acquired infection, re-infection by a seropositive donor, or reactivation of latent recipient viruses [5].

The most common of these virus infections is CMV, a double-stranded DNA virus of the Herpesviridae family with a seroprevalence rate of 70–90% in the general adult population [6]. The dormant forms of CMV are usually resident in subpopulations of CD34^+^ myeloid progenitor cells as well as in CD14^+^ lymphocytes, monocytes, dendritic cells, and megakaryocytes [6]. Reactivation of latent viruses occurs when there is a reduction in cellular immune activity (especially involving CD8+ cells) and also due to certain cytokines such as tumour necrosis factor-alpha (TNF-α) and Interleukin 1β (IL-1β), that trigger latent virus transformation [6]. Characteristic laboratory results include leukopenia, thrombocytopenia and raised liver enzymes. In severe cases, CMV viremia can progress to tissue-invasive disease, resulting in pneumonitis, gastroenteritis, retinitis, meningitis or colitis [5]. In addition, cell-signaling proteins such as cytokines, chemokines and growth factors released during CMV infection lead to further impaired immune response. Consequently, there is increased susceptibility to other opportunistic infections, graft rejection and in the longer-term, malignancy [7]. Overall, the greatest risk factor for post-transplant CMV viremia is donor-recipient serology mismatch, i.e., CMV seronegative patients receiving a kidney allograft from a CMV seropositive donor (D+R−) [8]. The incidence of CMV infection in this group of patients is estimated to be around 60% [9].

EBV, a double-stranded DNA gamma herpesvirus, is another ubiquitous pathogen seroprevalent in more than 90% of the adult population. It persists within B lymphocytes without active infection in the majority of hosts. In transplant recipients, however, newly acquired or reactivated latent infections can cause invasive diseases. The spectrum of such diseases range from non-neoplastic asymptomatic viral replication through EBV end-organ diseases (pneumonia, encephalitis/myelitis, and hepatitis) to EBV-mediated post-transplant lymphoproliferative disorder (PTLD) [10]. There is no current standardized treatment for patients who become EBV positive, but remission is routinely achieved through the reduction of immunosuppression with or without additional antiviral therapy [11].

Until recently, polyomavirus infections were not recognized as common sequelae of transplantation. However, rates of infection in renal transplant patients have increased in recent decades with the advent of newer and more potent immunosuppressive agents. Incidence rates now range from 10–60% [12]. The two polyomaviruses, BKV and JCV, are double-stranded DNA viruses of the Papovaviridae family with widely differing clinical manifestations [13]. After primary infection in childhood, they remain latent in the renal tubular epithelial cells but can be reactivated in states of relative or absolute cellular immunodeficiency. When reactivated, BKV replication occurs in the kidney tissue resulting in viruria, viremia, and sometimes rapid renal cell lysis [14]. BK virus infection can manifest as hemorrhagic and non-hemorrhagic cystitis, ureteric stenosis, and nephritis. However, most infections in transplant recipients occur without clinical signs except for a rise in serum creatinine [3] [NO_PRINTED_FORM]. BKV nephropathy has been reported to cause graft loss in up to 24% of affected kidney transplant recipients within 2 years [15,16]. In contrast to BKV, reactivation of JCV may lead to dissemination of the virus to the nervous system where it infects and replicates in oligodendrocytes leading to progressive multifocal leukoencephalopathy (PMR) a demyelinating disease of the central nervous system [4]. JCV causes virus-associated nephropathy less frequently than BKV [17].

Recent advances in immunosuppressive therapy have reduced the incidence of acute rejection, but unfortunately, have worsened post-transplant infection rates [17], hence the need to understand the incidence, predictors, and consequences of these infections.

This study therefore aims to assess the incidence and risk factors and outcome of CMV, EBV, BKV and JCV infection in a large cohort of kidney transplant recipients transplanted and managed over a 20-year period.

## 2. Methodology

### 2.1. Study Population

In this retrospective cohort study, we collected data from 962 subjects who received kidney transplantation between 01 January 2000 and 31 December 2021 and who were followed at a tertiary nephrology centre, the Salford Royal Hospital, Northern Care Alliance NHS Foundation Trust (overall catchment population 1.55 million). Subjects are usually repatriated to our hospital after a period of typically 3–4 months following transplantation at the regional transplant centre. Repatriation of some recipients might be delayed beyond the fourth month post-transplantation if there were early post-transplant complications that required close monitoring such as severe acute rejection, ureteric stenosis, or allograft vascular abnormalities. We excluded subjects who suffered graft loss within 3 months post-transplantation.

### 2.2. Data Collection and Definitions

Demographic, clinical and laboratory data were extracted from the electronic patient records (EPR). The data were collected until one of the following clinical endpoints occurred: graft loss, death, loss to follow-up, or the end of the study period (31 December 2021) which ever occurred first. Infections were identified from virology reports identifying patients who had developed a positive polymerase chain reaction (PCR) test for CMV, EBV, BKV or JCV virus since the date of their latest transplantation.

Preimplantation transplant factors included preemptive transplantation, donor type, Human leucocyte antigen (HLA) mismatch and donor and recipient CMV status. perioperative factors included cold ischemia time and post-operative factors such as immunosuppression regimen, history of acute rejection, post-transplant cardiovascular events malignancy and baseline estimated glomerular filtration rate (eGFR) (measured at 3 months post-transplantation) were also recorded. eGFR was calculated using the Modification of Diet in Renal Disease (MDRD) Study equation.

Post-transplant outcomes comprised of death censored graft survival (DCGS), dialysis-free survival (DFS) and death with a functioning allograft.

The overall rate of change in eGFR (delta eGFR) was compared between the four viral infection groups as well as between patients with any post-transplant viral infection (infection group), and patients without any post-transplant viral infection (non-infection group).

Death-censored graft loss (DCGL) was defined as a return to dialysis or re-transplantation. DFS was defined as a composite of DCGL and death with functioning graft.

Cardiovascular disease was defined as a composite of myocardial infarction, coronary artery disease, stroke, heart failure or peripheral vascular disease. Heart failure was defined as the clinical symptoms of ventricular dysfunction such as dyspnea and raised jugular venous pressure and/or bibasilar crackles corroborated by elevated natriuretic peptide levels and clinical or radiological evidence of pulmonary or systemic congestion [18]. PVD was defined as a disease of the upper and lower limb arteries arising from partial or complete obstruction due to arteriosclerotic changes [19]. Cardiovascular events were all adjudicated by the managing clinician and based upon hospital admission records from our own centre or outlying district hospitals. Data on CVD was retrieved from patients’ medical records.

Acute and chronic rejections were confirmed by kidney transplant biopsy (usually indicated by deterioration in graft function and worsening proteinuria).

All post-transplant malignancies were captured according to clinical history. Non-melanoma skin cancers were excluded from the study.

All subjects were treated with CNI-based maintenance immunosuppression as either tacrolimus or cyclosporin. Those without contraindications also received anti-proliferative immunosuppression, mycophenolic acid (MPA) or azathioprine (AZA). Those at high immunologic risk were also maintained on glucocorticoids.

We defined symptomatic infections as those that were associated with clinical signs and symptoms or laboratory evidence of organ dysfunction. In contrast, asymptomatic infections are those detected via routine viral screens without any associated clinical or other laboratory abnormalities.

For the individual viral strains, symptomatic CMV was defined as those with clinical or laboratory evidence of localized CMV infection (e.g., retinitis, gastrointestinal disease, endocrine disease, adenitis, nephropathy, pneumonitis), and/or disseminated disease (affecting multiple organ systems).

Symptomatic EBV was defined as the occurrence of mononucleosis or PTLD. Whereas symptomatic BK virus infection was defined as BK viremia associated with one or more of the following: haemorrhagic cystitis, ureteric stenosis or biopsy proven BK nephropathy, symptomatic JCV infection was defined as the occurrence of one or both of progressive multifocal leukoencephalopathy and JCV nephropathy.

BKV and JCV nephropathy was diagnosed if there was histologic evidence of polyoma-associated nephropathy following a clinically indicated kidney biopsy.

#### Viral PCR Surveillance and Assays

Monitoring for CMV, EBV, BK and JCV was usually undertaken monthly for the first 6 months or until repatriation from the regional transplant centre. Thereafter there was no routine policy for regular testing of patients during follow-up, but viral PCR testing was usually performed annually or if clinically indicated. More frequent testing to monitor viremia was conducted if an infection was confirmed.

Viral PCR testing was undertaken in one central virology laboratory using a TaqMan PCR-based approach. The CMV PCR assay was conducted using the real-time Taqman PCR assay for CMV which was first introduced in 1999 [20]. The CMV TaqMan PCR was designed to amplify the glycoprotein B gene of the CMV virus. The theoretical limit of sensitivity was 500 IU/mL of blood based on a sampling volume of 2 μL of EDTA blood [20]. The CMV TaqMan PCR assay was later redesigned in 2014 to target the DNA polymerase gene following evidence that this gene is more conserved in CMV gene pool. The limit of sensitivity is 500 IU/mL (Log 2.7).

Similarly, EBV real-time TaqMan PCR, using 100µL of patient blood and able to detect viral DNA over a linear span of between 100 and 10^7^ IU/mL [21]. However, the limit of sensitivity for the assay is 1000 IU/mL (log 3.0).

BK and JC polyomavirus assay was also carried out using the real-time TaqMan PCR which targeted the large T antigen (between 1999 and 2019). In 2019 it was redesigned to detect the VP2 gene of both BK and JC virus. The limit of sensitivity for BK and JC virus detection was 50 IU/mL.

All samples were processed with Qiagen MDX using a bespoke extraction which is suitable for all sample types, and amplification was carried out using Thermo 7500 real-time instruments. Assays were considered positive if test results showed a viral DNA count greater than or equal to the limit of sensitivity for the virus assayed.

### 2.3. Statistical Analysis

The cohort was first divided into 2 groups. The infection group consisted of subjects who had at least one episode of infection by any of the index viruses and the non-infection group consisted of subjects who experienced none of the evaluated infections. We also conducted a sub-analysis of individual DNA viral infections (CMV, EBV, BKV, JCV) to determine their incidence, time to infection, type of clinical disease, complications, treatment, and allograft outcomes. Continuous variables were presented as mean and standard deviation for normally distributed data or median and interquartile range for skewed data. For group comparison, the two-tailed *t*-test was used for parametric variables and Wilcoxon’s sign-rank test for non-parametric variables. Categorical variables were summarized as frequencies and percentages and compared using Pearson’s chi-squared or Fisher’s exact test.

For the evaluation of possible risk factors of DNA virus infection, multivariate logistic regression analysis was conducted. Covariates were chosen based on previous knowledge and if they satisfied the criteria for being a potential confounder [22]. The variables included in the multivariate logistic regression model include age at transplantation, gender, ethnicity, primary renal disease, type of CNI (cyclosporin vs. tacrolimus), number of immunosuppressive agents, history of acute rejection, baseline eGFR, smoking history, preemptive transplant, pre-transplant diabetes, donor type, donor, and recipient CMV serostatus, number of HLA mismatch, duration of pre-transplant dialysis and average post-transplant uPCR. The final variables in the model were then selected using a stepwise backward elimination process until only variables with a *p*-value less than 0.20 remained in the model.

Finally, two separate multivariate Cox-proportional hazards regression models were used to compare DCGS and DFS between recipients who developed viremia and those who did not and adjusted for the effects of confounders including age, gender, number of HLA mismatches, baseline eGFR, and history of biopsy-proven acute rejection (BPAR). Of those who experienced viremia, we estimated the probability of DCGL stratified by the virus subtypes using the Kaplan–Meier method. Statistical significance was defined as *p*-value < 0.05. All analysis were conducted using Stata version 14 statistical software package (StataCorp, College Station, TX, USA).

### 2.4. Ethics

The study complies with the declaration of Helsinki and was registered with the Research and Innovation department of the Northern Care Alliance NHS Group (Ref: S21HIP03) who approved the methodological protocol as outlined above. As this was a retrospective observational study based on routinely collected and fully anonymised data, the need for individual patient consent was waived by the Research and Innovation review committee, who granted study approval.

## 3. Results

### 3.1. Characteristics of the Study Cohort

The demographic and clinical characteristics of the cohort are summarised in Table 1. A total of 962 subjects were evaluated. Total follow-up was 7475 person-years (median 7.3 years per patient). The mean age at transplantation was 47.3 ± 15.2 years, 62% were male and 82% were of white ethnicity. Glomerulonephritis (27.4%) was the most common cause of ESKD followed by cystic kidney disease (13.8%) and diabetic nephropathy (12.4%).

30% of recipients (288) had at least one DNA virus infection (viremia group). There was no significant difference in age, sex, ethnicity, BMI or primary kidney disease between the viremia and non-viremia groups (Table 1). However, recipients who had viremia were more likely to have received a cadaveric donor allograft (75% vs. 24%), more likely to have a lower baseline eGFR at 3 months (45 vs. 51 mL/min/1.73 m^2^), and to have a higher average uPCR in the first year (20 vs. 15 mg/mmol). Furthermore, recipients with viremia had a significantly lower average haemoglobin concentration over the period of follow-up (123 vs. 126 g/L) and were more likely to be on long-term steroid therapy than those who did not experience viremia. However, we suspect that this association may have been a consequence of viremia rather than the cause, as viral infections are frequently managed by discontinuation of antimetabolites (MMF) and replaced with corticosteroids. The low haemoglobin and raised uPCR may have been consequences of viremia and viremia-induced loss of allograft function.

### 3.2. Infection Rates, Disease Complications and Management of Individual DNA Virus Infections

Table 2 shows the incidence, severity, clinical features and treatments offered for each of the individual DNA viral infections. It also shows the impact of the viral log count on disease severity. 14% (133/962) of recipients experienced CMV viremia with 33/133 (26%) experiencing symptomatic organ dysfunction. EBV viremia was diagnosed in 11% (109/962) of recipients of which 8 recipients (7%) progressed to PTLD. 9% (86/962) experienced BKV viremia of which 6 (7%) had BKV nephropathy; there was no BKV-associated hemorrhagic cystitis or ureteric stenosis. JCV viremia was diagnosed in 4.4% (42/962) of recipients with 2/42 (5%) developing JCV nephropathy. No case of JCV-associated progressive multifocal leukoencephalopathy (PML) was found. The median time to viremia post-transplantation was shortest in CMV, 9 months, and longest in EBV, 45 months. BKV and JCV infection occurred at a median time of 13- and 15-months post-transplant, respectively.

As shown in Table 2, more than half of the recipients who experienced CMV viremia underwent antiviral treatment in addition to immunosuppression reduction. In contrast only about 1% of those diagnosed with EBV or BKV infections were given antiviral treatment. None of the JCV infections received antiviral treatment. The mean log10 viral concentration was significantly higher in those with CMV viremia who developed symptomatic organ dysfunction than those with asymptomatic infections (mean log_10_ 4.6 vs. 3.1 IU/mL; *p* < 0.001) whereas EBV, BKV and JCV viral counts were not different between symptomatic and asymptomatic infections. Furthermore, those with higher viremia log values were more likely to receive escalated interventions during follow-up, notably a combination of immunosuppression reduction and antiviral agents (Table 2).

Figure 1 shows the effect of donor and recipient CMV sero-status on the incidence of CMV viremia post-transplant.

Recipients who developed CMV viremia were significantly more likely to have received their transplant from a CMV positive donor irrespective of their status whereas recipients of CMV negative donor allografts had a lower risk of CMV viremia. Only 7% of CMV viremia occurred in D−R− recipients; this increased to 16% in D−R+ recipients and further rose to 38% in CMVD+R+ recipients and 39% in CMVD−R+ recipients. This suggests that most CMV viremia occurred in recipients of CMV donor positive sero-status.

#### Risk Factors of DNA Virus Infection

Sixteen variables were included in the initial multivariate logistic regression model. By stepwise backward elimination of covariates with *p* > 0.20, six variables were left in the final model, as shown in Figure 2. The factors associated with lower odds of viremia are male recipients (OR 0.65; 95% C1: 0.44–0.98; *p* = 0.037), and higher baseline eGFR, (OR 0.87; 95% CI: 0.79–0.95; *p* = 0.003) per 10 mL/min increase. Conversely factors associated with increased odds of viremia include CMVD+R- serostatus (OR 2.40; 95% CI: 1.32–4.35; *p* = 0.004), CMVD+R+ (OR 1.95; 95% CI: 1.14–3.34; *p* = 0.015), Total HLA mismatch (OR, 1.15; 95% CI: 1.01–1.32; *p* = 0.041), cyclosporin as CNI rather than tacrolimus (OR 2.70; 95% CI: 1.10–6.65; *p* = 0.030) and number of maintenance immunosuppressive medications (OR 1.62; 95% CI: 1.13–2.30; *p* = 0.008) for each additional agent.

### 3.3. Transplant and Recipient Outcomes

Over the follow-up period, 87 out of 962 recipients (9%) lost their graft and 161/962 (17%) died with functioning grafts. The most common causes of graft loss included chronic allograft injury (CAI) (38%), acute rejection (24%), and recurrent glomerulonephritis (14%).

Regarding post-transplant complications, post-transplant cardiovascular disease was more likely to occur in recipients who experienced viremia (26% vs. 20%; *p* = 0.042). Post-transplant malignancy was also more common in recipients who had viremia than in the non-viremia group (11% vs. 7% *p* = 0.14). Incidence of acute rejection, and new-onset diabetes mellitus after transplantation (NODAT) did not differ between the groups (Table 1). There were nine cases of PTLD in the cohort, eight of these occurred in recipients who had EBV viremia post-transplant.

Figure 3 shows the relationship between DNA virus infection and the slope of eGFR decline. Recipients with a history of viremia had a statistically significant average annual drop in eGFR compared with those with no history of post-transplant viremia (−1.41 ± 4.09 vs. –0.64 ± 4.61 mL/min/1.73 m^2^/year; *p* = 0.0232). Individually, recipients who had CMV, EBV, BKV and JCV viremia all showed a steeper rate of eGFR loss with JCV having the greatest decline (2.40 mL/min/1.73 m^2^/year), followed by CMV (1.50 mL/min/1.73 m^2^/year), EBV (1.36 mL/min/1.73 m^2^/year) and BKV (1.19 mL/min/1.73 m^2^/year), Figure 3A.

The eGFR slopes became steeper as the number of infection types increased (shown in Figure 3B) suggesting a cumulative impact of multiple different types of viral infections on eGFR decline. Patients who had a history of JCV had the highest incidence of multiple viral infections (46%) followed by CMV (42%), EBV (40%) and BKV (37%) (Table 2). Interestingly this was mirrored by the rate of GFR decline (Figure 3A).

### 3.4. Outcomes of Asymptomatic Compared to Symptomatic Infections

For each of the virus strains, recipients with symptomatic infections showed a higher rate of eGFR decline compared to those with asymptomatic infections. Mean slope of eGFR decline in symptomatic vs. asymptomatic patients was −1.89 vs. −0.87 mL/min/year for CMV, −3.66 vs. −1.83 mL/min/year for EBV, −2.13 vs. −1.73 mL/min/year for BKV and −3.47 vs. −3.17 mL/min/year for JCV. Dialysis-free survival was also lower for recipients who experienced symptomatic viral infections compared to those who were asymptomatic. For instance, for recipients with symptomatic vs. asymptomatic CMV infections, this was 12.8 vs. 13.9 years. For recipients who experienced EBV infection, dialysis-free survival was 11 years for symptomatic and 15 years for asymptomatic infections. A similar result was noted for BKV (9.7 vs. 15.6 years) and JCV (2.6 vs. 14.7 years).

**Figure 3 viruses-14-02406-f003:**
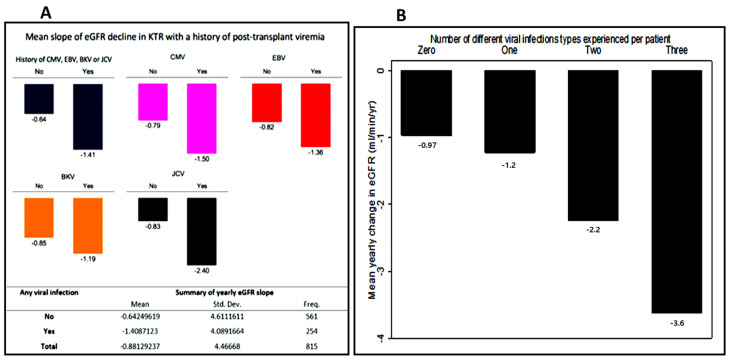
Mean yearly eGFR change (delta eGFR) of patients who had viremia compared to those with no history of viremia. (**A**) Mean yearly eGFR change (delta eGFR) of patients who had viremia compared to those with no history of viremia. (**B**) mean delta eGFR stratified by the number of different viral infections experienced.

In terms of the effect of viremia on graft outcomes, we found that recipients who experienced viral infections were more likely to experience DCGL. Although only CMV was associated with increased DCGL risk when individual viruses were analyzed (Figure 4). In the multivariate Cox regression model shown in Figure 5a,b, death-censored graft loss adjusted for age, gender, total HLA mismatch, baseline eGFR and history of acute rejection was significantly higher in patients with a history of viremia than those without (adjusted hazard ratio aHR 1.74; 95% C1: 1.08–2.80; *p* = 0.024) (Figure 5a). However, dialysis- free survival adjusted for the above confounders did not differ between the two groups (aHR 1.13; 95% C1: 0.86–1.47; *p* = 0.038) (Figure 5b).

## 4. Discussion

In this retrospective single-center cohort study of 962 kidney transplant recipients, we explored the factors associated with post-transplant DNA viremia and the prognostic impact of experiencing these infections on allograft and recipient outcomes. 30% of kidney allograft recipients in our cohort experienced at least one episode of CMV, EBV, BKV or JCV infection over the period of follow-up. A quarter (71/288) of those with a history of viremia had more than one type of viral infection. Male gender and higher baseline creatinine were associated with a lower risk of viremia whereas pretransplant donor-positive CMV serology (irrespective of recipient CMV status), cyclosporin maintenance therapy and high immunosuppression burden were associated with the development of viremia. CMV viremia was the most commonly occurring of the four DNA viral infections, followed by EBV and polyomaviruses. The time to infection post-transplant was shortest with CMV infection (9 months) and longest with EBV (45 months). A quarter of patients with CMV infections experienced symptomatic organ dysfunction of which gastroenteritis was the most common feature. Less than 10% of patients with EBV (7%) or polyoma viremia (6%) suffered symptomatic disease. Symptomatic CMV viremia was associated with higher viral DNA count compared to asymptomatic infection (log 5.4 vs. 4.0) *p* < 0.001 whereas EBV, BK and JCV viral counts did not differ between complicated and asymptomatic infections. Our study revealed a higher rate of eGFR decline in recipients with a history of viremia compared to those with no history of viremia. Recipients who experienced more than one type of viremia were also more likely to have a more rapid loss of eGFR, and a greater incidence of post-transplant cardiovascular disease and malignancy. Death-censored graft loss adjusted for confounders was significantly higher in recipients with a history of viremia than those without but there was no difference in dialysis-free survival between the 2 groups.

A similar study by McCaffrey et al. evaluated the same four viral infections in a small cohort (98 patients) of paediatric transplant patients and reported a two-fold higher overall incidence of viremia, 67.3% [17,23]. The difference in the result is likely due to the population studied. Children are more likely to be sero-naïve to these viruses, pre-transplantation and will therefore be at higher risk of primary infection post-transplantation. The incidence of DNA viral infections in a study of 561 adult KTR by Hwang et al. was 34.2%, similar to our findings [23].

The reported incidence of individual virus types differed amongst previous studies due to differing surveillance practices, duration of follow up and sensitivity of the assay. Furthermore, prospective studies tend to report a higher incidence of viremia than retrospective studies most likely due to a more proactive (research-oriented) surveillance procedure. Previous studies estimated the incidence of CMV in kidney transplant populations to be between 8% and 32% [12] which is in line with the findings of this study. The incidence of EBV viremia in one prospective active surveillance study was found to be as high as 40% in the first year of transplantation alone [24]. This is much higher than the findings of this study, suggesting that a significant number of patients may have experienced asymptomatic undiagnosed EBV infections. Like EBV, the reported incidence of polyomaviruses (BKV and JCV) in renal transplant populations are 10–45% most of these being asymptomatic infections [25].

As has been reported previously [12], it was not surprising to find that the burden of immunosuppression is a strong predictor of DNA viremia. Reactivation of latent infection occurs because of impaired cellular immunity due to post-transplant immunosuppression, especially the use of lymphocyte-depleting agents [6]. Furthermore, as expected, the burden of immunosuppression tends to be higher in recipients with a greater number of HLA mismatches. This may explain the higher likelihood of viremia with increasing number of HLA mismatches.

According to our results, recipients of CMV donor-positive allografts are at greater risk of contracting viremia than those who received CMV donor-negative allografts. This is consistent with findings of the study by McCaffrey et al. [17] and shows that receiving a CMV positive allograft not only predisposes to CMV infection (due to reactivation of latent virus) but also increases the risk of infection from other DNA viruses as well. The explanation for this can be derived from evidence that CMV infection increases susceptibility to other infections by down-regulating the expression of HLA antigens on T-cells and antigen-presenting cells [26]. CMV can also inhibit the proliferation of T-cells and reduce their ability to secrete IL-2 and interferon-gamma. CMV also disrupts the opsonization of infected cells, reduces complement binding and disrupts macrophage cytoskeleton and migration. Consequently, both the innate and adaptive immune responses to other infections are further compromised, resulting in increased susceptibility to other infections [26].

There is a paucity of evidence on the impact of gender on the risk of viremia in transplant recipients. However, evidence from studies on immunocompetent populations suggests significant differences between the sexes in their immune responses to infections. This was attributed to the effects of sex hormones on females as well as the extra X chromosome in females [27,28]. Compared to their male counterparts, immunocompetent adult females display better local viral clearance and enhanced innate and adaptive immune responses [27]. This suggests that men are more vulnerable to viral infections and some bacterial infections, while women are more susceptible to autoimmune diseases and have higher immunoreactivity toward pathogens [28]. Hence, we were surprised to observe an association between male gender and lower odds of viremia. There was a similar finding in Yalci et al. who found that women KTR infection rates were higher than that of men (61% vs. 46.5%) *p* = 0.045 [29] One explanation for this is the fact that the dialysis-free survival in our cohort was significantly shorter in men than in their female counterparts, so women were exposed to the risk of viral infections for longer periods than men.

Our finding of decreased odds of infection with higher eGFR broadly supports the widely known fact that lower kidney function is associated with poorer immune response [30,31]. Low kidney function has been shown to impair both the innate and the adaptive immune response. This is due to decreased B and T lymphocyte numbers, poor lymphocyte activation, impaired monocyte function, inadequate antigen presentation, weakened memory cell generation and insufficient antibody production [32].

Unlike some previous studies [33,34], we did not find any association between likelihood of viremia and age at transplantation, or with acute rejection.

Our study found that in all four DNA virus groups, viremia was associated with higher average annual decline in eGFR when compared to the non-viremia groups (Figure 3A). This is further supported by the finding that recipients who suffered multiple types of infection had a steeper decline in eGFR with a steeper eGFR slope as the number of infection types per recipient increased (Figure 3B). These results suggest that even viral infections with low viral counts may not be prognostically benign and they underline the importance of prevention, regular surveillance, and monitoring of recipients at-risk of viral infections.

Pre-transplant screening enables the risk stratification of patients especially as regards CMV and EBV viremia. Prophylactic antiviral agents (e.g., valganciclovir) for CMV are commonly given to at risk individuals for the first 3 months after transplantation in most transplant centers. These are usually recipients with CMV D+R− serostatus and those who received a lymphocyte depleting agent at immunosuppression induction. Based on the result of this study, a case could be made for extending CMV prophylaxis to include all recipients of CMV donor positive allografts (CMV D+R− and CMV D+R+). The duration of antiviral prophylaxis aims to strike a balance between prevention of viremia and development of antiviral resistant strains. Most centers offer 3 months of prophylaxis. However, considering that the median time to CMV infection in our cohort was nine months, increasing the duration of prophylaxis is a worthy consideration.

A significant finding from this study was that viremia adversely affected death-censored graft survival. Previous studies have reported conflicting results about the impact of viremia on graft outcomes. For instance, Blazquez-Navarro et al., noted that patients with viremia had significantly lower GFR in the first-year post-transplant compared to those with no viremia [3]. On the contrary, McCaffrey et al. found no difference in the rate of graft failure or death between those who had viremia and those who did not. [17]. There are reports linking CMV infection to the development of chronic allograft rejection, transplant glomerulopathy and atherosclerotic renovascular disease [12]. The effect of viruses on allograft survival requires further prospective studies.

A major strength of this study was the relatively large number of study subjects investigated, the detailed analysis of relevant baseline characteristics as well as post-transplant outcomes. Nevertheless, there are limitations that need to be acknowledged, first of which is the retrospective nature of the study. Secondly, viral surveillance beyond six months post-transplantation was inconsistent and depended on the supervising physician. This meant that asymptomatic and transient infections may have been undiagnosed and unrecorded. As a result, the true prevalence of viremia may have been under-represented by the observed prevalence. Thirdly, for some of the subjects, data from the regional transplant center where they were followed for the first 3–6 months post-transplantation were incomplete or unavailable. For these subjects, data on early post-transplant viremia was not available for analysis. Despite these limitations, our study evaluated the incidence of four key viral infections in an adult KTR population cohort. As far as we know, these infections have not been evaluated together previously.

## 5. Conclusions

In conclusion, one third of our cohort of kidney transplant recipients had a history of post-transplant DNA viremia. Donor CMV positive serostatus, the type and burden of immunosuppression, female gender and low baseline allograft function were independent predictors of risk of viremia. Viremia was associated with a decline in allograft function and a higher risk of graft loss. In addition, GFR loss was faster when patients experienced multiple different types of viral infection. These findings highlight the need for adequate pre-transplant risk stratification, optimal prevention strategies, effective surveillance and appropriate intervention including those in recipients with low viral counts.

## Figures and Tables

**Figure 1 viruses-14-02406-f001:**
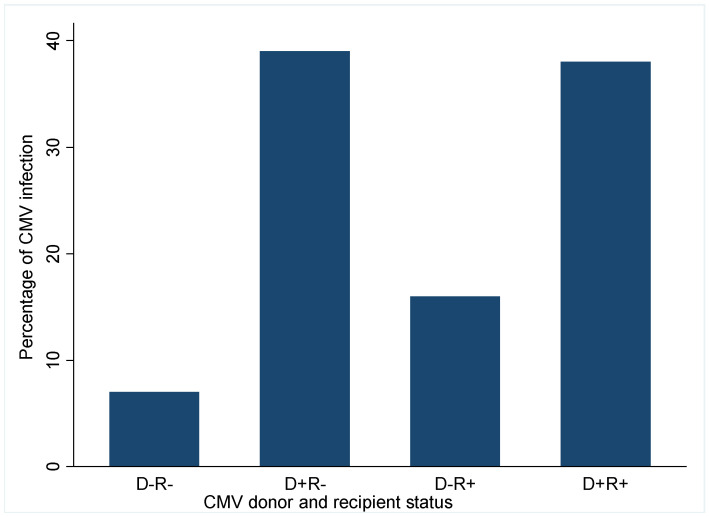
Impact of Donor and recipient CMV serostatus on incidence of CMV viremia. D, donor; R, recipient; +, positive; −, negative.

**Figure 2 viruses-14-02406-f002:**
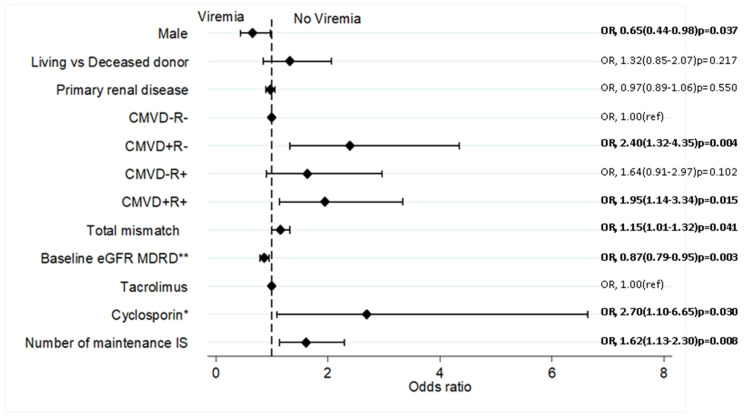
Independent predictors of DNA virus infections by Multivariate analysis. DNA virus infections is a composite of CMV, EBV, BK and KC virus post-transplant infections. Covariates were chosen based on previous knowledge; multivariate logistic regression conducted by stepwise backward elimination of covariates with *p*-values ≥0.20; ** per 10 mL/min/1.73 m^2^; * compared to Tacrolimus; IS, immunosuppression; MDRD, modification of diet in renal disease equation; OR, Odd ratio; ref, reference variable.

**Figure 4 viruses-14-02406-f004:**
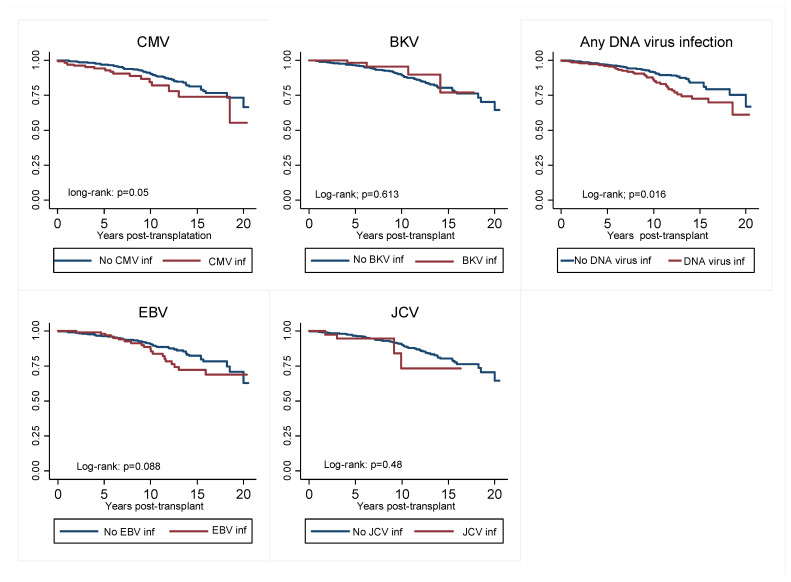
Kaplan–Meier curve showing graft survival censored for death of viremia group compared to non-viremia group for individual DNA virus infections, as well as those who experienced any viremia vs. those who did not. CMV, cytomegalovirus; EBV, Epstein Bar virus; JCV, John Cunningham.

**Figure 5 viruses-14-02406-f005:**
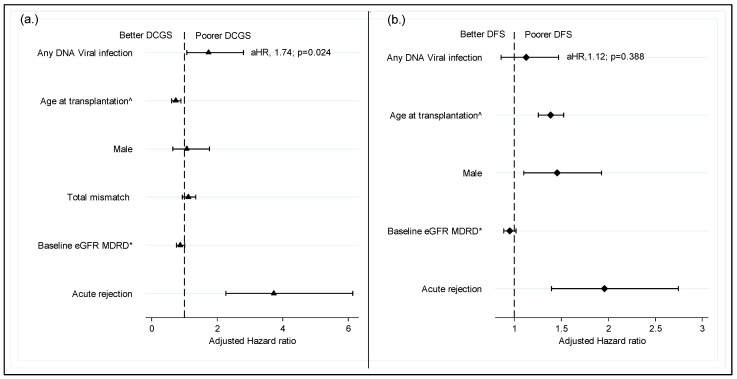
**Impact of DNA virus infection on graft survival censored for death**. (**a**) Impact of DNA virus infection on graft survival censored for death; (**b**) dialysis free survival. ^ per decade of age; * Per 10 mL/min/1.73 m^2^; DCGS, death censored graft survival; DFS, dialysis free survival (a composite of graft loss and death with functioning graft; MDRD, modification on diet in renal disease; aHR, adjusted hazard ratio.

**Table 1 viruses-14-02406-t001:** Demographic and clinical characteristics of infection group vs. the non-infection group.

Variable	*Total* *n = 962*	*No Viremia* *n = 674*	*Viremia* *n = 288*	*p-Value*
**Demographics**				
Mean age at Tx (years) ± SD	47.3 ± 15.2	46.9 ± 15.6	48.3 ± 14.1	0.279
Median age at Tx (years)	48 (3–82)	48 (3–82)	48 (13–79)	0.615
Male, *n* (%) Female, *n* (%)	596 (62) 365 (38)	419 (62.5) 251 (37.5)	177 (60.8) 114 (39.2)	0.119
White, *n* (%) Black, *n* (%) Asian, *n* (%) Other, *n* (%)	784 (81.4) 19 (2.0) 134 (13.9) 24 (2.5)	551 (82.2) 9 (1.4) 91 (13.6) 19 (2.8)	233 (80.1) 10 (3.4) 43 (14.8) 5 (1.7)	0.748
BMI, *n* ± SD	26.7 (12.5–72.3)	26.8 ± 5.3	26.7 ± 6.2	0.197
Smoking history ^α^, *n* (%)	279 (34.6)	191 (34.5)	88 (34.7)	0.946
**Primary renal disease**				
Glomerulonephritis, *n* (%)	268 (27.8)	180(26.7)	88 (30.4)	0.447
Diabetic nephropathy, *n* (%)	121 (12.6)	85 (12.6)	36 (12.5)	
Chronic hypertension, *n* (%)	68 (7.0)	45 (6.7)	23 (8.0)	
Cystic kidney disease, *n* (%)	126 (13.1)	83 (12.3)	43 (14.8)	
Unknown, *n* (%)	142 (14.8)	102 (15.1)	40 (13.8)	
Reflux disease	137 (14.2)	104 (15.5)	33 (11.4)	
Others, *n* (%)	100 (21.4)	74 (11.0)	26 (9.0)	
**Pre-transplant comorbidities**				
DM, *n* (%)	167 (17.4)	115 (17.0)	52 (18.2)	0.652
CVD, *n* (%)	214 (22.2)	143 (21.2)	71 (24.5)	0.256
MI, *n* (%)	38 (4.0)	28 (4.2)	10 (3.4)	0.587
CHF, *n* (%)	36 (3.7)	24 (3.6)	12 (4.1)	0.684
PVD, *n* (%)	57 (5.9)	38 (5.7)	20 (6.9)	0.472
CVA/TIA, *n* (%)	56 (5.8)	37 (5.5)	19 (6.5)	0.54
Liver disease, *n* (%)	15 (1.6)	11 (1.6)	4 (1.4)	0.758
Solid tumours, *n* (%)	57 (5.9)	42 (6.3)	15 (5.2)	0.501
Haematological malignancies, *n* (%)	9 (0.9)	6 (0.9)	3 (1.0)	0.841
**Total number of transplants**				
1, *n* (%)	842 (87.6)	585 (87.3)	257 (88.3)	0.254
2, *n* (%)	104 (10.8)	77 (11.5)	27 (9.3)	
3, *n* (%)	11 (1.1)	5 (0.8)	6 (2.1)	
4, *n* (%)	4 (0.4)	3 (0.4)	1 (0.3)	
**Donor type**				
Live donor, *n* (%)	277 (29.0)	209 (31.0)	68 (23.5)	0.016
Cadaveric, *n* (%)	665 (69.1)	449 (66.7)	216 (74.7)	
Total mismatch (mean ± SD)	2.44 ± 1.42	2.40 ± 1.42	2.50 ± 1.40	0.348
**CMV status**				
D−R−	164 (23)	130 (26)	34 (16)	0.008
D+R−	140 (20)	86 (17)	54 (25)	
D−R+	163 (23)	114 (23)	49 (23)	
D+R+	250 (35)	173 (34)	77 (36)	
**Immunosuppression regimen**				
Tacrolimus, *n* (%)	835 (86.9)	593 (88.5)	242 (83.2)	0.119
Cyclosporin, *n* (%)	70 (7.3)	42 (6.3)	28 (9.6)	
Other, *n* (%)	5 (0.5)	5 (0.7)	1 (0.3)	
**Initial antimetabolite**				
MMF/Myfortic, *n* (%)	657 (68.3)	487 (72.7)	170 (58.4)	0.688
Azathioprine, *n* (%)	112 (11.7)	81 (12.1)	31 (10.7)	
**Steroid regime (prednisolone)**				
Steroid Avoided	414 (47.4)	313 (52)	101 (37.5)	<0.001
Maintenance > 6 m, *n* (%)	79 (9)	53 (9)	26 (10)	
Maintenance < 6 m, *n* (%)	379 (43)	237 (39)	142 (53)	
**Biochemical parameters**				
Baseline Creatinine (umol/L)	130 (107–160)	130(110–172)	136(105–154)	0.002
Baseline eGFR (mL/min/1.73 m^2^)	49 (5–90)	51 (5–90)	45 (5–90)	0.005
First year uPCR (mg/mmol)	16 (9–38)	15(8–36)	20 (10–43)	0.007
Average Hemoglobin (g/L) ^β^	126 ± 19	127 ± 19	123 ± 20	0.003
Total cholesterol (mmol/L) ^¥^	4.3 (3.6–4.9)	4.3 (3.6–4.9)	4.4 (3.7–4.9)	0.285
**Post-transplant outcomes**				
History of acute rejection, *n* (%)	104 (10.8)	66 (9.7)	38 (13.7)	0.069
History of NODAT, *n* (%)	157 (16.3)	119 (17.4)	38 (13.6)	0.156
Post-transplant CVD, *n* (%)	205 (21.3)	134 (19.6)	71 (25.5)	0.042
Post-transplant malignancy ^µ^, *n* (%)	81 (8.4)	48 (7.0)	33 (11.4)	0.014
Graft loss, *n* (%)	87 (9.0)	52 (7.6)	35 (12.6)	0.014
Death, *n* (%)	161 (16.7)	110 (16.0)	51 (18.3)	0.394

Continuous variables: Presented as mean ± SD or median (interquartile range) and compared using the independent *t*-test test or Wilcoxon’s sign-rank test. Categorical variables: Presented as number (percentage) and compared using Pearson’s chi-squared test. BMI, body mass index; Ca^2+^, corrected calcium; CHF, congestive heart failure; CVA/TIA, cerebrovascular accident/transient ischaemic attack; CVD, cardiovascular disease; DM, diabetes mellitus; eGFR, estimated glomerular filtration rate; HbA1c, haemoglobin A1C (glycated haemoglobin); MI, myocardial infarction; PVD, peripheral vascular disease; Tx, transplant; ^α^ Defined as current or ex-smoker., ^β^ average over the follow up period; ^¥^ mean total cholesterol in the first year of transplantation; ^µ^ excluding non-melanoma skin cancer.

**Table 2 viruses-14-02406-t002:** Incidence, severity, clinical manifestations and treatments of the individual DNA viral infections.

	CMV	EBV	BKV	JCV
Incidence	133 (13.8)	109 (11.3)	86 (8.9)	42 (4.4)
Median time to infection(m)	9 (5–39)	45 (23–82)	13 (8–25)	15 (8–42)
Viral PCR log median (IQR)	3.37 (2.8–4.5)	3.0 (3.0–3.53)	3.52 (2.8–4.9)	3.28 (2.65–4.17)
Viral PCR log range	(2.08–6.81)	(3.0–5.88)	(1.69–6.94)	(1.71–6.66)
More than 1 type of viremia	56 (42)	44 (40)	32 (37)	19 (46)
**Clinical Features**				
Asymptomatic infection	93 (70)	100 (92)	80 (93)	40 (95)
Symptomatic infection	33 (25)	8 (7)	6 (7)	2 (5)
Unknown	7 (5)	1 (1)	N/A	N/A
**Disease type**				
Nephropathy	2 (1.6)	0 (0)	5 (7)	2 (4.8)
Pneumonitis	6 (4.8)	N/A	N/A	N/A
enteritis	18 (14.3)	N/A	N/A	N/A
Meningitis/retinitis	2 (1.6)	N/A	N/A	N/A
Adenitis	1 (0.8)	0 (0)	N/A	N/A
Disseminated disease	3 (2.4)	N/A	N/A	N/A
PTLD	N/A	8 (7.3)	N/A	N/A
Ureteric stenosis	N/A	N/A	0 (0)	0 (0)
Haemorrhagic cystitis	N/A	N/A	0 (0)	N/A
PMR	N/A	N/A	N/A	0 (0)
other	1 (0.8)	0 (0)	0 (0)	0 (0)
**Treatment**				
No intervention	29 (22.6)	87 (81.31)	26 (36.6)	15 (53.6)
Immunosuppression reduction only	26 (20.3)	19 (17.76)	42 (59.2)	13 (46.4)
Immunosuppression reduction plus antivirals	73 (57) ^α^	1 (0.93) ^β^	1 (1.4) ^π^	0 (0)
Immunosuppression reduction plus other treatment		0 (0)	2 (2.8) ^Ω^	0 (0)
**Viral count Log value stratified by symptoms and by treatment [IU/mL (median (IQR)]**				
Viral count median(range)	3.48 (2.08–6.81)	3.015 (3.00–5.88)	3.52 (1.69–6.94)	3.28 (1.71–6.66)
Asymptomatic patients	3.1 (2.8–4.0) *p* < 0.001	3.03 (3.0–3.5) *p* = 0.102	3.5 (2.79–4.89) *p* = 0.29	3.2 (2.7–4.1) *p* = 0.16
Symptomatic patients	4.6 (3.7–5.4) *p* < 0.001	3.0 (3.0–3.0) *p* = 0.102	4.2 (4.19–4.98) *p* = 0.29	4.4 (4.1–4.6) *p* = 0.16
No treatment given	2.7 (2.7–3.01) *p* < 0.001	3.0 (3.0–3.34) *p* < 0.001	2.78 (2.12–3.50) *p* = 0.002	2.89 (2.25–3.24)
Immunosuppression reduced	2.93 (2.77–3.74) *p* < 0.001	3.69 (3.33–4.17) *p* < 0.001	4.21 (3.16–5.26) *p* = 0.002	3.24 (2.65–4.17) *p* = 0.0049
Immunosuppression + Antiviral/other treatment	4.2 (3.47–5.17) *p* < 0.001	_	4.94 (1.7–4.98) *p* = 0.002	_

^α^ Ganciclovir or valganciclovir; ^β^ Acyclovir; ^π^ Cidofovir; ^Ω^ IV immunoglobulin; CMV, Cytomegalovirus; EBV, Epstein Bar virus; JCV. N/A, Not applicable; PTLD, Post-transplant lymphoproliferative disease; PMR, progressive multifocal leukoencephalopathy.

## Data Availability

The datasets used and/or analysed during the current study are available from the corresponding author on reasonable request and with permission of the research and innovation department of Northern care alliance NHS Foundation Trust.

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
