# Peer review of "Kidney Transplant-Associated Viral Infection Rates and Outcomes in a Single-Centre Cohort"

_viruses, 2022, doi:10.3390/v14112406_

Round 1

Reviewer 1 Report (Previous Reviewer 1)

Revision of the manuscript has resulted in some improvements. However, I still have concerns about the basic study design.

1) Overall, PCR testing for viremia was not standardized and seemed to show significant gaps post repatriation. Thus, the true prevalence of viremia for different DNA viruses remains incompletely understood.

2) It is highly doubtful whether a single low level positive PCR read during follow-up has the same clinical significance as a sustained high level PCR read or as viral activation noted in a patient with apparent DNA-virus induced disease. Such distinctions are not made by the authors (see lines 242-251: the categories of "asymptomatic" and "symptomatic" infections have to be clearly defined for each viral strain and results analyzed in much greater detail.

3) Some outcome analyses are rather ill-defined, such as "heart failure" or "peripheral vascular disease" (line 136). Most likely such adverse events are driven by high blood pressure rather than viral activation simply detected at some juncture by PCR.

4) Minor aspect: ref#4 seems to be ill chosen. Replace with PMID: 32654412 and PMID: 29279304. The referenced BK-virus associated graft failure rate appears to be too high. Line 241: The detected positivity for BKV and JCV infections by PCR in plasma seems to be surprisingly late post transplantation; this is possibly a reflection of the study design and time points selected for PCR testing.

5) Minor point: lines 237/238 - how was JCV (vs BKV) nephropathy diagnosed?

Author Response

Reviewer 2 Report (Previous Reviewer 2)

In my opinion, the revised statistical approach greatly improves the quality of the manuscript.

Minor comments:

Line 248: shouldn't "P<0.001" be within the parenthesis?

Figure 2: In my understanding, "No Viremia" should be placed on the left side and "Viremia" on the right side.

Line 315: shouldn't this read "graft loss" instead of "graft survival"?

Lines 385-392: The authors might also consider an effect of ganciclovir treatment on the bone marrow and hence immune function.

Author Response

Reviewer 3 Report (Previous Reviewer 3)

The authors meet the comments of the comments that I have suggested. The paper can now be accepted.

Author Response

Thank you for reviewing our article. We are grateful for the comments and delighted to hear that the paper can now be accepted.

This manuscript is a resubmission of an earlier submission. The following is a list of the peer review reports and author responses from that submission.

Round 1

Reviewer 1 Report

The authors are studying a large cohort of kidney transplant recipients and focus on patients with some evidence of viral infections based on PCR test results. They correlate findings with selected outcome criteria.

I have concerns about the study design and conclusions drawn.

1) No information is provided regarding PCR analyses. How often were patients tested, did a positive PCR test result trigger more frequent analyses and/or anti-viral treatment when appropriate (such as with CMV infections)? Was testing done in one central laboratory or in different laboratories with different testing modalities/assays? What exactly was tested, i.e. plasma, serum, whole blood? 

2) DNAemia of "DNA-Viruses" can be transient and self limiting. The chosen cut-offs with a single PCR test result of >500 copies for CMV or >250 copies for BKPyV or JCPyV are very arbitrarily set and very low. It is, indeed, questionable whether low level DNAemia carries any independent prognostic significance. 

3) There is evidence that in several viral infections -other than CMV- a seropositive donor/seronegative recipient status predisposes for post transplant viral infection/activation. This aspect was not studied here.

4) Table 5: Post transplant adverse events are listed here. However, no information is provided how these events were evaluated, e.g. acute rejection confirmed by indication biopsy, protocol biopsy or based on clinical parameters only. Chronic rejection remained undetermined. How was CVD assessed etc etc. How is post transplant malignancy defined, are minor tumors such as BCCs included or excluded?

5) How did pre-existing conditions influence post transplant outcome? Would the exclusion of patients with major preexisting conditions change the outcome analysis?

Reviewer 2 Report

Infections caused by opportunistic viruses remain an important complication after organ transplantation, although prophylaxis and treatment regimens are available for some of these viruses. Medication for prophylaxis or treatment may have adverse effects, and detection of viral infections may result in adjustment of the immunosuppressive regimen, which in turn may increase the risk of graft failure. In this context, the manuscript by Kalra and colleagues is valuable information as it compares the frequency of four relevant opportunistic viruses (HCMV, EBV, BKV, JCV) in adult kidney transplant recipients and attempts to identify risk factors for these infections and the impact of these infections on patient outcomes.
Although data on infection with these viruses in renal transplant recipients are already available, the data presented here are novel in that a systematic comparison of these four viruses in adult renal transplant recipients in one study has not been published previously. Importantly, the results differ from similar data obtained in a pediatric kidney transplant cohort and are therefore not merely additional evidence.

My main criticism of the manuscript is that the authors limit their statistical analysis to univariate analyses of the association of "infection" with potential risk factors and outcomes, whereas I miss an attempt to analyze in more detail the interdependence of the various factors they associate with "infection."

For example, prolonged corticosteroid treatment was associated with infection, and infection was associated with certain outcomes (cardiovascular disease, malignancy, and graft loss), raising the question of whether these outcomes were primarily related to the duration of corticosteroid treatment or whether infection was an independent risk factor for any of these outcomes. Without knowing the details of the data sets to which the authors have access, I assume that a multivariable regression analysis is feasible. At the very least, they should consult an expert in biomathematics and either add an appropriate analysis or explain why this was not possible.

Minor issues:

in line 80 the authors report the finding that "the biggest risk factor for PTLD is having no previous exposure to EBV, which is most frequently the case in younger populations", which would be clearer mentioning that this applies only if the donor is EBV-positive.

The layout of tables 1, 4 and 5 is slightly off and should be adjusted in the final version.

In lines 297-299, the authors might consider adding the information that, consistent with their explanation, EBV and not CMV was the most common infection in the padiatric cohort. This is easily explained by the fact that most CMV infections already occur during breastfeeding, whereas the incidence of EBV infections in later childhood and adolescence is much higher compared with CMV infections.

In Tab. 6, I do not understand what these data tell about the correlation of "age at Tx" and death and/or graft loss. What does a hazard ratio of 1.029... mean? And what does this finding mean regarding the role of infections in the posttransplant setting?

Reviewer 3 Report

In this paper, the authors have retrospectively examined a cohort of 961 KTR transplanted, enrolled between 2000-2021. They were grouped into those that experienced a viral infection (CMV, EBV, BKV, JCV) and those without a history of viral infections, and then clinical parameters were examined. The authors reached the conclusion that post-kidney transplant viral infections were associated with a higher risk of cardiovascular disease and malignancy but not with reduced dialysis-free survival. Corticosteroid maintenance therapy was associated with a higher risk of post-transplant viremia.

Overall, the analyses seemed to be run correctly. Even though the topic is deeply investigated in the literature, lacking a bit of novelty, a major strength of the work is the large sample size and the detailed statistical analysis. The manuscript is well-written and organized. The figures are adequate and are of help to follow the manuscript.

Some minor improvements are required to allow the publication of the work:

- Figure 1. Since the paper analyzes not only CMV, but also other viruses, Figure 1 should include all the viruses discussed in the paper.

- Tables seem to be written in a font different compared to that used throughout the manuscript.

-         -  Figure 2, legend: remove capital letters if they are not necessary.

-          - Graphs should be uniform (fonts, background, style, etc).

Round 2

Reviewer 1 Report

None